# OpenReview forum: "Task Success is not Enough: Investigating the Use of Video-Language Models as Behavior Critics for Catching Undesirable Agent Behaviors"
_colmweb.org/COLM/2024/Conference — COLM_

### Official Review · Reviewer_eCKj · 2024-04-26

**Rating:** 6
**Confidence:** 5
**Ethics Flag:** 1

**Summary:**

This paper explores a crucial aspect of robotic agents’ performance -- ensuring that while they achieve task goals, they also adhere to a broader range of constraints and preferences that go beyond mere task completion. To this end, the authors seek to improve the performance of robots in daily tasks by using large vision and language models (VLMs) to catch their undesirable behaviours and then refine them.

**Questions To Authors:**

1. Besides recall and precision, are there any other metrics or methods used to evaluate the effectiveness of the VLMs in identifying undesirable behaviors?
2. Can the authors discuss how the proposed system performs across various types of robots and different operational environments? What steps are taken to ensure the system’s robustness and generalization?
3. What specific challenges do the proposed method face that result in lower precision after adding grounding part?
4. How are the critiques generated by the VLMs integrated back into the robotic systems to improve behavior? How to ensure the critiques from VLMs are correct? What mechanisms are in place to ensure that these critiques lead to meaningful behavioral adjustments?

**Reasons To Accept:**

1. The paper introduces a novel use of video-language models (VLMs) as behavior critics in the context of robotics.
2. The proposed benchmark consisting of diverse scenarios where robots achieve goals but may exhibit undesirable behaviors is valuable. It provides a structured way to assess and improve robot behavior, offering a tool that other researchers can use to benchmark their systems.

**Reasons To Reject:**

1. Precision issues. The VLMs demonstrate lower precision after incorporating visual grounding. This indicates a potential reliability issue in the critiques provided by the models, which could lead to incorrect assessments of robot behavior.
2. Generalization concerns. The paper does not explicitly address how well the proposed system generalizes across different types of robots or environments.
3. Dependence on quality of training data. The effectiveness of the VLMs heavily depends on the quality and variety of the training data. Any biases or limitations in the video dataset might lead to suboptimal behavior critiques, impacting the overall system performance.
4. Lack of integration details. More information might be needed on how the behavior critiques are integrated back into the robotic systems for improving behavior.

---

> ### Author Rebuttal · Authors · 2024-05-30
>
> We thank reviewer eCKj for valuable comments. Here are the clarifications on some of the comments and raised questions:
>
> >### Precision issues even after incorporating visual grounding
>
> Please note that, as shown in Table 1, adding visual-grounding feedback **increases**  the precision rate of GPT-4V from ~62% to ~98%. If reviewer eCKj meant to inquire about the mild decrease in recall rate, an explanation can be found at the bottom of page 8.
>
> >### Metrics other than recall and precision
>
> We presented a suite of metrics to evaluate the feasibility and show the practical challenges of applying VLMs as critics to guide future research. Overall, we have:
> 1. Second paragraph of Sec. 5.1: Accuracy of determining the presence of undesirability (evaluated with both positive and negative samples)
> 2. Recall & Precision
> 3. Sec. 5.2: Insights according to a taxonomy of error modes
> 4. First paragraph of Sec. 5.4 & Table 3 & Appx. A.1: accuracy of preference labels when asked to compare video pairs.
> 5. Appx. A3: In the CaP demos, we reported the number of iterations & verbal critiques needed to produce satisfactory policies.
>
> >### Generalization concerns (across different types of robots or environments)
>
> When constructing the benchmark, we have made substantial efforts to ensure it covers a diverse range of common household tasks and environments (e.g., in different rooms or with different styles of furniture). Also, the problem of criticizing videos is closer to event recognition, which typically is not as tightly tied to any specific robot morphology and kinematics compared to control or policy learning problems. So the type of robot here may not be a primary factor affecting generalization. However, we agree that for a more comprehensive benchmark, increasing diversity in this dimension (i.e., types of robots) could be beneficial. We are continuously working on expanding our benchmark in this regard.
>
> >### More information of how the behavior critiques are integrated back to the robotic system
>
> In the paper, we present a practical closed-loop system with Code-as-Policies (CaP) as the robot controller (last paragraphs of Sec. 4.3 & Sec. 5.4 & Appx. A.3). For feedback integration, verbal critiques are listed in follow-up messages within the policy synthesis conversation (with the LLM-based CaP agent). In response, the CaP agent will re-generate a policy. Due to space constraints, we moved some implementation details to Section A.3 in the appendix.

---

> > ### Comment · Reviewer_eCKj · 2024-06-05
> >
> > The rebuttal has addressed my concerns and I am happy to keep my positive rating.

---

### Official Review · Reviewer_habn · 2024-05-06

**Rating:** 8
**Confidence:** 3
**Ethics Flag:** 1

**Summary:**

### Problem addressed
When employing large-scale generative models in robotics and embodied AI,
despite their capability to generate meaningful solutions, they frequently fall
short of producing desired outputs on the first try. To address this challenge,
the paper investigates the potential of utilizing large vision and language
models (VLMs) as scalable Behavior Critics for identifying undesirable robot
behaviors in videos.

### Main contributions
A comprehensive evaluation of VLM critics in real-world household environments,
showcasing their strengths and failure modes, and providing guidelines on their
effective utilization in an iterative process of policy refinement.

### Methodology
- VLMs as behavior critics: Frames from a video for a policy for a
  short-horizon manipulation task (GPT-4V and Gemini Pro) are provided along
  with an instruction prompt that asks the VLM to identify any undesirable,
  potentially harmful, or risky behaviors in the video.
- Benchmarking VLMs behavior critics and analysis of failure modes: 114 test
  cases are prepared, each containing 2 negative videos and 1 positive video of
  different policies attempting a task, along with human annotations. A taxonomy
  of 4 types of errors is proposed to study the failure modes of the VLM
  critics.
  - In addition, it is experimented to have a critic compare pairs of videos and
    provide pairwise preference.
- Augmenting behavior critics with grounding feedback: Grounding feedbacks
  calling out hallucinated events are provided to the VLM critic to improve the
  accuracy of the critics, which significantly improves the precision rate.
- Integrating critiques into a closed-loop system: A viable closed-loop system
  is demonstrated, where Code-as-Policies (CaP) is employed to control the
  robot, receiving the feedback from a VLM critic regarding undesirable
  behaviors, and accordingly adjust its policy.

### General evaluation
- Quality: The methodology, experiments, and analysis are well-conceived and
  well-presented.
- Clarity: The paper is well-written and easy to follow, with clear explanations
  of methods and logic.
- Originality: The work is practical yet innovative.
- Significance: This paper presents a solid and valuable study on the
  interesting application of using VLMs as critics evaluating robotics policies
  with respect to constraints that are hard to quantitatively and exhaustively
  delineate. The direction is promising given the fast advancement of VLM
  models, and holds the potential of boosting the field of embodied AI
  tremendously.

**Questions To Authors:**

1. Will the curated benchmarking dataset be released?

2. Could authors provide more detailed statistics of the collected data? For
  example, concrete task names and the number of suboptimal / optimal policy
  videos collected and annotated for each task.

3. How exactly are the 114 test cases, each containing 3 videos, constructed with
   the 175 videos? Apparently there are videos reused. Could such reuse introduce
   biases to the evaluation metrics? This also makes the detailed data statistics
   more necessary to report.
   1. A related question, on page 8, in section 5.2, it is mentioned that there
       are 262 critiques. Are these 262 critiques made on all 175 videos? What is
       the distribution of number-of-critiques-per-video?

4. The taxonomy seems to be focused on false alarms. Could the errors where a VLM
  critic *misses* undesirable behaviors (measured by recall) be divided to
  categories and analyzed as well?

5. On page 5, in section 4.2, the "Evaluation metrics" part says

   > “precision” should permit the critics to suggest valid critiques even when they do not associate with any human-annotated negative event.

   while on page 8, in section 5.1, the second paragraph says
   > Note that the recall-precision evaluation is conducted solely with videos containing negative events

   It is now unclear to me whether "positive videos" are considered when evaluating precision.

6. Could the "critique operationalizability" be improved by prompt engineering or
  few-shot in-context examples? When the VLM critic can correctly identify an
  undesirable behavior, it seems possible that its critiques can be more
  operationalizable if guided / prompted accordingly.

7. Why is the taxonomy designed such that the error types are not mutually
  exclusive? Is the "causal understanding errors" vs "visual grounding errors"
  the only overlapping case?

**Reasons To Accept:**

- The topic is very interesting and relevant, and the paper is a solid piece of
  work that establishes a framework for evaluating and understanding VLMs as
  policy critics.

**Reasons To Reject:**

- The benchmarking dataset could be further expanded both in size as well as in
  the coverage of scenarios.
- The taxonomy proposed is not fully justified.

---

> ### Author Rebuttal · Authors · 2024-05-30
>
> We thank reviewer habn for valuable comments. Here are the clarifications on some of the comments and raised questions:
>
> >### Evaluation metrics, particularly recall and precision
>
> Reviewer 1 (reviewer 2XFC) asked a similar question, please refer to our response (the first response) there. Note that there we referred to “positive videos” as “appropriate videos”, and “negative videos” as “inappropriate videos”.
>
> >### Whether the dataset will be released & more detailed statistics of the collected data & expanding the dataset
>
> Yes, we plan to release the dataset once the reviewing period ends & we are continuously expanding the dataset based on new undesirable cases observed.
>
> Regarding the construction of 114 test cases with 175 videos, yes, some videos are reused. Specifically, one positive video may be reused for multiple negative videos. Such reuse doesn’t introduce any significant bias to our evaluation, because we examine and report the behavior of VLM on negative videos and positive videos separately. We do acknowledge that a more balanced and expanded dataset would be ideal. But please note that, considering the cost of mining undesirable behaviors and collecting videos, the primary objective of this research would be to show the feasibility of this new direction such that the community sees the value and could offer joint efforts to make the dataset more complete and be usable for other purposes (e.g., for fine-tuning) – similar to how the Open X-Embodiment dataset was constructed jointly.
>
> We will add a section in Appendix to provide detailed stats in camera ready. For now, we have 51 different instructions (tasks), each with at least two negative videos and one positive video. Note that this already covers a large fraction of tasks in the largest public robotics datasets like BRIDGE and Open-X. We also plan to list all the instructions in the camera ready.
>
> >### Are these 262 critiques made on all 175 videos?
>
> As mentioned in the first sentence of Sec. 5.2, the analysis there focuses on negative videos. In most cases, one negative video features one negative behavior. GPT-4V outputs around 2 critiques on avg per response (which lowers the precision rate).
>
> >### Why the error types are not mutually exclusive?
>
> Yes, causality error + visual grounding error is the only overlapping case in our experiment. We believe it is more sensible to allow this when the VLM claims a non-existing event as the cause of a negative outcome.

---

> > ### Comment · Reviewer_habn · 2024-06-04
> >
> > I would like to thank the authors for their response, which answered several of my questions, while not addressing some other of my questions (#4, #6).
> >
> > I am still be interested to hear more about the unanswered questions, and I do have some follow-ups regarding the answered ones:
> >
> > - Question 5: Thanks for the clarification in the response to Reviewer 2XFC. If I understand correctly, the positive videos are then not used for calculating the reported precision / recall. In that case, I would suggest the authors to revise the sentence on page 5, in section 4.2, which sounds to me suggests that the precision's calculation actually takes into account the positive videos.
> >   > “precision” should permit the critics to suggest valid critiques even when they do not associate with any human-annotated negative event.
> >
> > - Question 7: I understand that the "causal understanding errors" and "visual grounding errors" conceptually do not have to be mutually exclusive, but having overlap would lead to complications in following analysis, and in fact without revealing the overlap count, the reported information is incomplete. I would argue that, it might be worthwhile to break down the "causal understanding errors" into two parts, one due to grounding errors, the other due to the VLM making wrong connections.

---

> > ### Author Response · Authors · 2024-06-06
> >
> > We thank reviewer habn for valuable comments. Please note that we didn’t answer some questions because of the 2500 character limit. We are happy to discuss them more here.
> >
> > >### Revising the sentence on page 5, in section 4.2
> >
> > We thank the reviewer for the constructive feedback. We will revise the sentence to: `"precision" should permit the critics to expand the set of human-annotated negative events`.
> >
> > We also plan to revise the first sentence in the same paragraph to: `Recall and precision are fundamental metrics in assessing behavior critics **on videos that showcase suboptimal behaviors**.`
> >
> > >### Revealing the overlap count of "causal understanding errors" and "visual grounding errors"
> >
> > The number of overlapping cases is 12 for GPT-4V without grounding feedback. These cases account for nearly 50% of the "causal understanding errors" (25 cases) but make up only a small fraction of the "visual grounding errors" (116 cases).
> >
> > We agree with the reviewer’s suggestion to break down the “causal understanding error”, and we will revise the paper accordingly.
> >
> > >### Q4. The taxonomy focuses on false alarms.
> >
> > In general, “missing undesirable behaviors” means the VLM didn’t generate criticisms for some undesirable events in negative videos. This leaves very limited space for us to analyze the “failure modes” (in contrast, in the case of false alarms, the VLM at least outputs something, which gives us more space to investigate the discrepancy between the inaccurate output and the “ground truth”).
> >
> > If the reviewer is interested in or inquiring about the patterns of the occurrence of “missing undesirable behaviors” errors, we did conduct some preliminary examination and didn’t observe any clear pattern. Such errors occur “evenly” across the test samples (with different task types and environments). The occurrence is generally unpredictable – the VLM may succeed in one test case while miss a similar negative event in another test case with minor variations (e.g., same type of manipulation task but with a different object, such as orange -> eggplant, or on a different table). We can include a discussion on this in the final version.
> >
> > Another reason why we focused more on false alarms is because false alarms could mislead/disrupt policy generation more compared to “missing undesirable behaviors” (although the latter means humans need to spend more time in supervising policy generation/learning).
> >
> > >### Q6. Could the "critique operationalizability" be improved by prompt engineering or few-shot in-context examples?
> >
> > We experimented with asking follow-up questions (with in-context examples) to GPT-4V to get more concrete & actionable critiques. We found that, although GPT became more verbose, its responses were not significantly more actionable compared to the original critiques. After all, giving operationalizable critiques sometimes can be as hard as solving the plan generation problem.
> >
> > Please note that we have chosen to leave this for future research and therefore haven’t included these findings in the current version. We believe more comprehensive experiments are needed before drawing conclusions. We would also like to note that this may be out of the scope of this study as we have put substantial resources into dataset construction, manual/human evaluation of VLM critiques and real-robot experiments.

---

> > > ### Comment · Reviewer_habn · 2024-06-06
> > >
> > > Thank you very much for the further response sharing insightful findings. My positive opinion of this work remains unchanged, and I'll maintain my rating as 8.

---

### Official Review · Reviewer_rUo6 · 2024-05-08

**Rating:** 6
**Confidence:** 3
**Ethics Flag:** 1

**Summary:**

Generative models often overlook task constraints and user preferences, and instead focus solely on achieving task success

To address this, the authors propose using video-language models (VLMs) as behavior critics to catch undesirable agent behaviors in videos.

The authors conduct a study on using VLMs to evaluate robot behaviors and find that they can identify a significant portion of undesirable behaviors.

The authors also demonstrate a framework for integrating VLM critiques into a closed-loop policy-generation system, which can refine policies according to the critiques.

**Reasons To Accept:**

The framework includes a benchmark dataset of 175 videos, a set of evaluation metrics. This framework provides a robust and systematic way to evaluate the performance of VLMs in this context.

This approach addresses the limitations of VLMs in visual grounding and provides a way to refine the outputs of VLM critics. The authors also demonstrate the effectiveness of this approach in achieving a high precision rate.

 The authors discuss the potential of integrating VLM critiques with closed-loop policy-generation systems, such as Code-as-Policies.

**Reasons To Reject:**

The evaluation is conducted on a relatively small dataset, which may not capture the full range of possible errors or scenarios.

The experiment relies on perfect grounding feedback to improve the performance of the VLM critic. However, in real-world scenarios, feedback may be imperfect, random and noisy. Also, how does the VLM critic handle hazzard during robot execution?

The experiment focuses primarily on GPT-4V and Gemini-Pro. I would like to know more results about other (weaker) VLMs' (e.g. Instruct GPT, Kosmos, Llava) performance in order to justify the contribution of the proposed methodology.

---

> ### Author Rebuttal · Authors · 2024-05-30
>
> We thank reviewer rUo6 for valuable comments. Here are the clarifications on some of the comments and raised questions:
>
> >### The evaluation is conducted on a relatively small dataset
>
> We agree that further expanding the dataset can be beneficial, and we are continuously working on this. Nonetheless, we want to note that, considering the cost of mining undesirable behaviors and collecting videos, the primary objective of this research would be to show the feasibility of this new direction such that the community can see the value and could offer joint efforts to make the dataset more complete and be usable for other purposes (e.g., for fine-tuning) – similar to how the Open X-Embodiment dataset was constructed jointly. Moreover, we have prioritized the diversity of suboptimalities and scenarios when collecting our dataset. This is reflected in the number of different tasks (over 50) within our 175 videos (note that this already covers a large fraction of tasks in the largest public robotics datasets like BRIDGE and Open-X).
>
> >### Handling of grounding feedback & weaker language models
>
> The main purpose of the grounding-feedback experiment is to investigate the ideal conditions under which the GPT-4V critic can have improved critique precision. The observation can guide follow-up research. We understand that handling noisy feedback and trying weaker VLMs are practical problems that need attention, but these may be out of the scope of this study as the objective of this work is to show the feasibility of a new research direction (also given that we have put substantial resources into dataset construction, manual/human evaluation of VLM critiques and real-robot experiments). Moreover, several papers we cited have specifically sought to tackle the noisy feedback problem (e.g., the Socratic Models). In the camera-ready, we will discuss future steps more and mention these practical challenges.

---

### Official Review · Reviewer_2XFC · 2024-05-12

**Rating:** 6
**Confidence:** 4
**Ethics Flag:** 1

**Summary:**

The paper addresses the automatic verification of robot behaviors.
This paper investigates using large vision and language models (VLMs) for automated verification of robot behavior. The authors empirically evaluate the ability of VLMs to detect undesirable robot actions given short videos and prompts. They find that GPT-4V, the VLM they use, tends to produce false positives (incorrectly identifying desirable actions as undesirable) when evaluating manipulation videos in which the grabber tool was primarily controlled by humans. In contrast, Gemini Pro tends to produce false negatives (missing inappropriate actions).

**Questions To Authors:**

* Why were Recall and Precision evaluated using data that includes only undesirable behaviors?
* What is the estimated frequency of undesirable behaviors in real-world applications? While the occurrence rate depends on the robot's policy, some assumptions are necessary for designing evaluation methods.
* Was the prompt template tuned for Gemini Pro? The readers may wonder whether prompt templates exist that could improve Gemini Pro's Recall.

**Reasons To Accept:**

* The application of VLMs to detect inappropriate robot behaviors is an interesting and promising area of research with the potential to enhance real-world robot applications.
* The paper identifies the main challenge, Visual Grounding, in using current VLMs to detect undesirable behaviors. Furthermore, it demonstrates that humans can address this challenge to improve detection accuracy, providing insights into potential limitations and guiding future research.
* The proposed method is experimentally evaluated on a real robot's behavior policy.

**Reasons To Reject:**

* The authors evaluate Recall and Precision using data that includes only inappropriate behaviors, potentially overestimating Precision. Since practical robot policies should minimize undesirable actions, evaluating these metrics on data that includes appropriate behaviors is essential to predict real-world performance. Additionally, Table 1 presents Recall and Precision for data with undesirable behaviors alongside Accuracy for data with both appropriate and inappropriate behaviors, which could be misleading.
* The lack of data availability hinders future research.
* The experimental results indicate a significant gap between the demonstrated accuracy and realistic applications. The authors should consider limiting the environment and tasks to more focused scenarios with higher applicability, given the diverse nature of the household tasks considered in the study.

---

> ### Author Rebuttal · Authors · 2024-05-30
>
> We thank reviewer 2XFC for the valuable comments. Here are clarifications on some of the comments and raised questions:
>
> >### Clarification on evaluation metrics (accuracy & recall & precision)
>
> At the top level, there is a binary classification problem where the VLM needs to determine the presence of any undesirable events in the video. For this aspect, we evaluated the VLM using both "appropriate videos" and "inappropriate videos", and reported the Accuracy.
>
> However, the binary classification results alone do not provide a complete picture. For “**inappropriate videos**,” it is crucial to evaluate whether the VLM's verbal critiques are valid - that is, whether it criticizes the videos for the correct reasons. This is why we report recall (the fraction of “ground truth” bad behaviors that are successfully identified by the VLM) and precision (the fraction of valid critiques within all generated critiques). Recall and precision in this context have their specific definitions, and they assess a more specific aspect than the general accuracy metric above (as explained in the subsection Evaluation Metrics under Sec. 4.2).
>
> For any critiques generated for “**appropriate behaviors**”— where no undesirable event is present —most of such critiques are invalid or “hallucinated,” so it would be unnecessary to further look into their “recall/precision”.
>
> We acknowledge that presenting “accuracy” alongside our specific “recall/precision” in the same table could lead to some confusion. We plan to separate them in the camera-ready once we have more space.
>
> >### Task applicability & frequency of undesirable behaviors
>
> We don’t think statistics on this are available or tractable to obtain. However, we would like to assure the reviewer that the tasks and the undesirable behaviors in our benchmark are of high applicability and practical utility. In fact, this research is motivated by tasks & undesirability that we observed in either public robot datasets (e.g., Bridge and Open X-Embodiment) or imperfect trajectories sampled from pre-trained policy models. In other words, scenarios in our test cases are widely recognized and considered within the community. Moreover, we believe people always want their robots to adhere to as many human preferences as possible in all tasks.
>
> >### The lack of data availability
>
> Reviewer 2 (reviewer rUo6) asked a relevant question, please refer to our response (the first response) there.

---

> > ### Comment · Area_Chair_7h7E · 2024-06-06
> > **Please see reviewer question**
> >
> > Hi Authors,
> >
> > You might have missed the questions from the reviewer. It would be great if you could respond.
> >
> > Thanks,
> > AC

---

> > ### Author Response · Authors · 2024-06-06
> >
> > Dear AC,
> >
> > We had our responses/rebuttal right under Reviewer 2XFC's comments. Could you please let us know if we have missed anything specific?
> >
> > We believe the only unanswered question was regarding the prompt template for Gemini Pro. We didn’t answer this in our initial rebuttal due to the 2500-character limit, but we are happy to provide a clarification here. Please note that, **at the time of this COLM submission, gemini-pro-vision (Gemini 1.0 Pro Vision) was the only widely released multi-modal model in the Gemini family**. Its capability is not comparable to GPT-4V and other newer & larger Gemini models such as Gemini 1.5 Pro and Gemini 1.0 Ultra (which is still not available through Gemini API). Hence, due to the model’s limited vision and reasoning capability, we find that Gemini-1.0-Pro-Vision did not offer improved performance even after careful tuning of the prompt template and in-context examples. We could provide some preliminary results from newer Gemini models at the time of camera ready if time permits. Besides, since Gemini-1.0-Pro-Vision is optimized for completion rather than chat/conservation, we also tried and made several prompt adaptations, such as in the ways of integrating grounding feedback (instead of providing feedback in follow-up messages as done for GPT-4V, we append the feedback as “auxiliary information” or as extra notes to the initial prompt). However, our results indicate these adaptations do not affect the output quality noticeably. All these findings suggest that the readers / future research should consider using the largest and latest Gemini models whenever possible. We can put a note regarding this in the revision.
> >
> > In addition to this, please feel free to let us know if there are any other explanations we can provide.
> >
> > Thank you.

---

### Decision · Program_Chairs · 2024-07-10

**Decision:**

Accept

**Comment:**

This is a very impressive idea - getting a vision-language model to act as a critic to assess the quality of robot behaviors, and the paper is well written and evaluated. The reviewers posed many questions, and the author responses were excellent, leading to great, detailed discussion. The key issues raised were handled well in rebuttal, and reviewers were satisfied with the response. A key reservation was the small dataset used for evaluation, however as the authors argue, this is the first study in a new area, and will hopefully provoke others to build upon the work - this seems a reasonable justification for the limited scale.